# A Novel Hypothesis on Choroideremia-Manifesting Female Carriers: Could *CHM* In-Frame Variants Exert a Dominant Negative Effect? A Case Report

**DOI:** 10.3390/genes13071268

**Published:** 2022-07-17

**Authors:** Niccolò Di Giosaffatte, Michele Valiante, Stefano Tricarico, Giulia Parise, Anna Maria De Negri, Guido Ricciotti, Lara Florean, Alessandro Paiardini, Irene Bottillo, Paola Grammatico

**Affiliations:** 1Division of Medical Genetics, Department of Experimental Medicine, San Camillo-Forlanini Hospital, Sapienza University, 00185 Rome, Italy; niccolo.digiosaffatte@uniroma1.it (N.D.G.); mvaliante@scamilloforlanini.rm.it (M.V.); giulia.parise@uniroma1.it (G.P.); lflorean@scamilloforlanini.rm.it (L.F.); paola.grammatico@uniroma1.it (P.G.); 2ASL Roma 2, 00100 Rome, Italy; stricarico@scamilloforlanini.rm.it; 3Neuro-Ophthalmology Clinic, San Camillo-Forlanini Hospital, 00152 Rome, Italy; adenegri@scamilloforlanini.rm.it; 4Department of Ophthalmology, University Hospital of Parma, 43126 Parma, Italy; guido.ricciotti@unipr.it; 5Department of Biochemical Sciences “A. Rossi Fanelli”, Sapienza University of Rome, Piazzale Aldo Moro 5, 00185 Rome, Italy; alessandro.paiardini@uniroma1.it; 6Division of Medical Genetics, Department of Molecular Medicine, San Camillo-Forlanini Hospital, Sapienza University, 00185 Rome, Italy

**Keywords:** Choroideremia, female carrier, X-linked disease, manifesting female, *CHM*, REP1

## Abstract

Choroideremia is an X-linked recessive condition presenting in males, with progressive degeneration of retinal and choroidal tissues leading to progressive visual loss. Its pathological mechanism is due to alterations in the *CHM* gene that encodes for REP1, a protein required for prenylation of Rab by the Rab geranylgeranyl transferase (RGGT). Even though female carriers are predicted to be not affected by the disease, a wide phenotypic spectrum ranging from mild to severe cases has been reported in women. The reason why Choroideremia manifests in female carriers remains elusive. While X chromosome inactivation (XCI) skewing has been proposed as a leading putative mechanism, emerging evidence has shown that *CHM* could variably escape from XCI. We described a family with an initial clinical suspicion of Retinitis Pigmentosa in which a novel *CHM* pathogenic splicing variant was found by exome sequencing. The variant, initially found in the 63-year-old female presenting with impaired visual acuity and severe retinal degeneration, segregated in the 31-year-old daughter and the 37-year-old son, both presenting with fundus anomalies. mRNA studies revealed a shorter in-frame *CHM* isoform lacking exon 10. Molecular modeling of the ternary REP1/Rab/RGGT protein complex predicted significant impairing of REP1/Rab binding without alteration of REP1/RGGT interaction. We suggest that, in our female cases, the biallelic expression of *CHM* may have led to the production of both the mutant and wild type REP1. The mutant isoform, sequestrating RGGT, could reduce its available amount for Rab prenylation, thus exerting a dominant-negative effect. If confirmed with further studies and in large cohorts of female carriers, the here proposed molecular mechanism could help to explain the complexity of manifestation of Choroideremia in females.

## 1. Introduction

Choroideremia (MIM #303100) is an X-linked recessive condition caused by pathogenic alterations in the *CHM* gene and characterized by the progressive degeneration of choroid, photoreceptors (PR) and retinal pigment epithelium (RPE). Its estimated incidence ranges between 1:50,000 and 1:100,000 worldwide [1]. Adolescent affected males develop night blindness, which is followed by a gradual loss of peripheral vision between the second and third decades of life and complete blindness in the later stages of disease [2]. Female *CHM* mutation carriers are mainly asymptomatic or mildly symptomatic [3], and only a minority of cases present a severe phenotype, consisting of retinal and choroidal atrophy such as that observed in affected males [4].

*CHM* contains 15 exons and is expressed in many tissues, including brain, spleen, endocrine tissues and eye, with the highest levels of transcription shown in the retina, choroid and RPE [3]. *CHM* codes for the Rab Escort protein 1 (REP1) [2], which plays a key role in the cytosolic prenylation (i.e., the addition of geranylgeranyl groups) to the Rab GTPases, allowing prenylated Rabs to anchor to the membrane of intracellular organelles and vesicles [5]. Indeed, REP1 allows the approach of soluble Rab with the Rab geranylgeranyl transferase (RabGGT, RGGT), which catalyzes Rab’s prenylation by using one or two molecules of geranylgeranyl diphosphate (GGDP) [5]. In the traditional proposed model of REP1-mediated prenylation, REP1 firstly binds to Rab and then escorts it to the RGGT-GGDP complex. After the formation of the REP1/Rab/RGGT ternary complex, RGGT mediates two cycles of Rab prenylation. The binding of a new GGDP molecule to RGGT mediates the detachment of the REP1/Rab-GG (Rab-geranylgeranyl) complex [6]. Interestingly, in 2008, Baron RA et Seabra MC demonstrated that in vivo Rab geranylgeranylation occurs preferentially via the pre-formed REP–RGGT-GGDP complex, so that a REP–RGGT pre-formed complex is needed for the binding of unprenylated Rab [7]. Traditional and alternative models of Rab prenylation are schematized in Figure 1. Alterations of REP1 cause a deficit of prenylation of different Rabs and, since these proteins are involved in regulation of intracellular membrane trafficking, different cellular processes are then impacted, finally leading to cell death.

A recent revision showed that at least 76% of the *CHM* reported alterations are gene/exon deletions, indels and nonsense variants, while splice-sites and missense changes, respectively, represent the 16% and 6% [3].

In this report, starting from the whole exome analysis of a family with retinal degeneration, we describe a *CHM* in-frame splicing variant leading to choroideremia manifestation in a male but also in two females. We then discuss a novel putative mechanism for the disease’s manifestation in female carriers.

## 2. Materials and Methods

### 2.1. Patients

The proband (Figure 2A, case I:2), a 63 years old woman, was referred to our hospital for genetic counselling with a suspicion of Retinitis Pigmentosa. In addition, the proband had suffered from hearing loss since she was 30 years old. Due to her personal clinical history, we initially assumed a possible diagnosis of Usher syndrome. A detailed family history revealed that case I:2 was the only member with hearing loss, while visual disturbances were reported for her 35-year-old son (patient II:1) and for her 32-year-old daughter (patient II:2). As for the other family members, there was no evidence of vision loss in the proband’s grandchildren, a girl and a boy aged, respectively, 5 and 3 years old (patient III:1 and III:2). During genetic counselling, the patients gave informed consent for the genetic analyses, which was approved by local ethic committees in accordance with the principles of the Declaration of Helsinki.

### 2.2. Ophthalmological Assessment

All of the patients underwent a comprehensive ophthalmological examination including best-corrected visual acuity (BCVA) measurement, Ishihara test for color deficiency, intraocular pressure evaluation, biomicroscopy of the anterior segment and fundoscopy after dilation. The patients also underwent the following diagnostic examination: color fundus photography, Optical Coherence Tomography (OCT), near-infrared reflectance (NIR) and Fundus Autofluorescence (FAF). Color fundus photographs of the posterior pole and the surrounding areas of both eyes were taken using the Digital Retinography System camera. (CenterVue SpA, Padua, Italy) OCT, NIR and FAF images were taken using A Spectral-Domain Optical Coherence Tomography (SD-OCT, Heidelberg Engineering, Heidelberg, Germany). In order to assess the degree of fundus involvement in female subjects, FAF images were compared with Edwards et al., 2015 [8] and classified accordingly.

### 2.3. Genetic Analyses

Whole Exome Sequencing (WES) was performed in case I:2 on genomic DNA extracted from peripheral blood. Target DNA regions were enriched by Nextera DNA Exome probes (Illumina, San Diego, CA, USA) and then sequenced on NextSeq550Dx sequencer (Illumina). Sequencing reads were aligned to the human reference genome (UCSC hg19) by BWA (v0.7.7-isis-1.0.2) (Illumina). Variant calling was performed by GATK Variant Caller (v1.6-23-gf0210b3). The DNA variants were annotated by Variant Interpreter (v.2.13.0.20) (Illumina). Variants’ mapping in genes associated to retinal disorders was prioritized and then filtered by MAF < 0.01 (GnomAD v2.1). Filtered variants were classified according to ACMG-AMP criteria [9]. The filtered variant was tested by Sanger sequencing both in case I:2 as well as in cases II:1, II:2, III:1 and III:2, with the following primers’ pair (annealing temperature: 56 °C, extension time: 30″):

*CHM* Ex10 FW: 5′-AGCCCTCAAAATAGCAACAAG-3′

*CHM* Ex10 Rv: 5′-CCCTAAAACCAGACCCTGTA-3′

To analyze the functional effect of the *CHM* splicing variant, mRNA from peripheral blood of case II:2 was retro-transcribed into cDNA and then sequenced by the Sanger technique with the following primers’ pair, spanning from *CHM* exon 8 to 13:

*CHM* Ex8-13 FW: 5′-CAATGACATCAGAGACAGCCA-3′

*CHM* Ex8-13Rv: 5′-TGTGCAAGTCAAATGAACCAA-3′

### 2.4. Molecular Modelling

The crystal structures of REP1 in complex with Rab geranylgeranyl transferase (GGTA; PDB: 1LTX [10]) and REP1 protein in complex with Rab7 (PDB: 1VG0 [11]) were used to model the human ternary complex of REP1/Rab7/GGTA, using internal functionalities of modeler-9 package [12]. PSI-BLAST was used to find suitable structural templates (percentage of identity > 85%) for the sequences to model [13]. The multiple alignment between the sequence to model and the templates was performed by using CLUSTALW [14]. Protein sequences analysis and visualization was performed using PyMod 3 [15]. The deficiency of *CHM* exon 10 was ab initio modeled using the official release of the neural network implemented in AlphaFold2 (version 2.0.1 [16]).

## 3. Results

### 3.1. Clinical Findings

On ophthalmological evaluation, patients I:2, II:1 and II:2 presented with retinal anomalies compatible with choroideremia.

*Patient I:2.* The proband presented complaining of blurred vision on distance and night blindness. The patient has a moderate myopia and BCVA was 0.63 in the right eye and 1.00 in the left eye. The patient presented a severe color defect at the Ishihara test. Anterior segment examination showed a bilateral lens opacification. Fundus examination showed a large bilateral chorioretinal atrophy, large peripheral pigmentary rearrangements and focal areas of Retinal Pigment Epithelium (RPE) disruption with light sparring of central macula pigmentation, especially in the left eye (Figure 3A,B). A SD-OCT was performed, showing extensive chorioretinal atrophy and retinal thinning in the right eye with hyper-transmission posterior to the RPE, while, in the left eye, a residual central island of preserved Ellipsoid Zone (EZ) area was present, with chorioretinal atrophy in the superior area of the macula. Pseudodendritic outer retinal tubulations (ORT) were present in the context of the chorioretinal atrophy in the left eye (Figure 3C,D). Areas of residual RPE could also be visualized in Infrared Reflectance (IR) images, mostly in the left eye compared to the right eye (Figure 3E,F). FAF images in both eyes showed generalized decreased autofluorescence, with residual areas of autofluorescence in the central macular area in the right eye with a male resembling pattern and also in the temporal macular area in the left eye with a geographic pattern (Figure 3G,H).

*Patient II:1.* The patient came to our department without complaining of any visual disturbance. His BCVA was 1.00 in both eyes. The patient presented a moderate color vision defect at the Ishihara test. The anterior segment of both eyes was normal. On fundus examination, large peripapillary and temporal retinal atrophy was found, with focal areas of RPE rearrangements in the peripheral retina and at the level of the vascular arcades. The foveal area appeared markedly pigmented in comparison with the surrounding area (Figure 4A,B). On SD-OCT, a large peripapillary chorioretinal atrophy with retinal thinning and hyper-transmission posterior to the RPE was revealed. In the nasal perifoveal region, small optically empty cysts were detected in the contest of the inner retinal layers. In the foveal and iuxtafoveal areas, the Ellipsoid zone (EZ), RPE and all retinal layers were well-detectable. Externally to the fovea, the RPE–Bruch’s complex appeared thinner, irregular and discontinuous and a new area of chorioretinal atrophy extended beyond the temporal macular region (Figure 4C,D). The transition of the intact RPE under the fovea to the atrophic area was well revealed with the IR images (Figure 4E,F). FAF images showed a male pattern with a central island of preserved autofluorescence (PAF) area (Figure 4G,H).

*Patient II:2.* The patient has mild myopia and her BCVA was 1.00 in both eye. No color vision defect was detected with the Ishihara test. Anterior segment examination was normal. On fundus examination, mild peripapillary atrophy was found, with areas of hyper-pigmentation in the perifoveal area. At the level of the vascular arcades, pigmentary changes were present, associated with areas of hyper-reflectivity. Macula seemed preserved (Figure 5A,B). On SD-OCT, the retinal profile was regular, with preservation of the physiological foveal depression. The inner retinal layer was preserved, but the Outer Nuclear Layer (ONL) and the Photoreceptor Layer were thinned in the perimacular zone. The Ellipsoid Zone (EZ) and the Interdigitation Zone (IZ) were irregular throughout all the macular and perimacular area. Even the RPE–Bruch’s membrane complex looked irregular, with small focal thickening points. Spots of hyper-transmission posterior to the RPE–Bruch’s membrane complex were present (Figure 5C,D). IR images revealed a normal aspect of the macular area and hypo-reflective areas among the vascular arcades and the mild periphery and focal hyper-reflective spot, corresponding to the focal thickening of the RPE (Figure 5E,F). FAF images showed a coarse pattern, with normal autofluorescence of the macular area and a generalized hypo-fluorescence of the vascular arcade area (Figure 5G,H).

*Patients III:1 and III:2.* Both patients had a natural visual acuity of 1.0 in both eyes and no defect in color vision on the Ishihara test. No significant changes were found on anterior segment and fundus examination.

### 3.2. Genetic Findings

By WES, case I:2 was found to carry the *CHM* (NM_000390.3):c.1349+1G>A variant in heterozygosity. The c.1349+1G>A maps at the donor site of *CHM* exon 10, and it is a rare variant, absent in the GnomAD dataset and not reported in the ClinVar database. Sanger sequencing also confirmed the presence of the variant in cases II:1, II;2, III;1 and III;2 (Figure 2A). The amplification of cDNA from case II:2, from *CHM* exon 8 to 13 and from subsequent analysis on agarose gel revealed the presence of two main signals of between 500 bp and 400 bp (Figure 2B). Sequencing analysis highlighted that the longer agarose band corresponded to a transcript including exons 8–13, while the shorter corresponded to the skipping of exon 10. Amplification of cDNA from a control showed the presence of only the longest band (Figure 2B). Exon 10 spans 105 nucleotides, so that its skipping due to the presence of the c.1349+1A allele should not alter the reading frame of *CHM* mRNA or create a truncated transcript.

The c.1349+1G>A was classified as a pathogenic variant by enabling the following ACMG/AMP_2015 criteria [9]: PM2 (absent from controls), PP1 (co-segregation with disease in multiple affected family members in a gene definitively known to cause the disease) and PS3 (functional studies (Section 3.3 below) supportive of a damaging effect on the gene splicing).

### 3.3. Molecular Modelling

To achieve structural insights on the potential effect of the c.1349+1G>A variant and exon 10 skipping in the mechanism of Rab GTPases geranylgeranylation carried out by the REP1/RGGT complex, we modeled the human ternary complex REP1 wild type and mutant/Rab7/RGGT using, as structural templates, the orthologous proteins from rattus norvegicus (PDB Codes: 1VG0 and 1LTX).

The wild type REP1/RGGT heterodimer shows a tight interface, which is formed by helices 8, 10 and 12 of the RGGT α subunit and helices D and E of REP1, hiding a small surface area of ~700 Å (Figure 6A). The RGGT/REP1 interaction is defined by conserved p-cation interactions, whose positions are also conserved in the c.1349+1G>A mutant. On the other hand, Rab7 interacts with REP1 via an extended interface involving the C-terminal domain, according to the crystal structures of the REP1 protein in association with Rab7 proteins. The C-terminus of REP1, where the peptide region codified by exon 10 is present, has a key role in coordinating such an interaction. The c.1349+1G>A mutation is responsible for the skipping of exon 10 and considerably alters the C-terminus of REP1 domain, losing the conformation that is committed for Rab binding (Figure 6B,C). Therefore, mutated REP1 would not be able to bind to Rab, yet is still able to interact with RGGT, sequestrating it and thus reducing the amount of RGGT available for geranylgeranylation of Rab proteins.

## 4. Discussion

We describe a family in which two adult females and an adult male, as well as two children, were carriers of the *CHM* c.1349+1G>A alteration, causing choroideremia in the adult cases. Genetic analyses demonstrated that the c.1349+1G>A led to the skipping of *CHM* exon 10 without the perturbation of the mRNA reading frame. Furthermore, by molecular modelling, the skipping of exon 10 was predicted to induce the expression of a REP1 protein with an alteration in the Rab-binding domain, but while maintaining its binding with RGGT.

Choroideremia is generally reported as an X-linked recessive condition, with female carriers generally being not affected or presenting with few clinical manifestations. However, in the here presented family, the two adult female heterozygous carriers (I:2 and II:2) of the c.1349+1G>A presented with significant degeneration of the fundus. Females presenting with such significant alteration of the retina are generally suspected of being affected by other conditions rather than Choroideremia, such as Retinitis Pigmentosa. Indeed, that was the initial clinical suspicion and the reason for referral of our proband (I:2). Nevertheless, the more the number of cases with *CHM* alterations increases in literature, the more it becomes clear that Choroideremia pathogenesis in females could be more complex than previously thought. Multiple attempts have been made to classify Choroideremia in female carriers and to clarify the wide spectrum of phenotypes. In 2015, by performing Fundus Autofluorescence examination and microperimetry studies on 12 *CHM* female carriers, Edwards et al. identified four FAF phenotype: fine, coarse, geographic and male pattern [8]. Moreover, a correspondent functional progressive impairment among these patterns was demonstrated by microperimetry [8]. More recently, Jauregui et al., based on reported symptoms and full ophthalmic examination (i.e., slit-lamp, dilated funduscopic examination, BCVA, short-wavelength fundus autofluorescence, spectral-domain optical coherence tomography and fundus color photography), classified a cohort of 12 female carriers into mild, intermediate and severe stages of disease [4]. As opposed to a static view of female carrier phenotypes, the authors proposed that manifestations in females should be treated as a continuum, with possible progression of retinal degeneration and worsening of the symptoms, albeit more slowly than in affected males [4].

Following the classification proposed by Edwards et al. [8], in the here reported family, case I:2 presented with a pattern resembling male FAF, while her daughter (case II:2) presented with a coarse FAF pattern. The difference between the mother and the daughter, both carriers of the same *CHM* variant, could indeed reflect two different age-related stages of Choroideremia, rather than two different stable phenotypes. Visual loss in Choroideremia is believed to occur by chance, as soon as the degeneration process includes the fovea [4]. In this view, the differences in the occurrence of blindness in males compared to females could reflect the greater chance of foveal involvement due to the fastest retinal degeneration occurring in males, because of their hemizygous state. Conversely, disease manifestation in female carriers should be caused by the reduction in *CHM* wild type allele expression or by the perturbation of wild type REP1 activity.

In X-linked recessive conditions, skewed X Chromosome Inactivation (XCI) is considered the main cause of disease manifestation in female carriers [17]. However, several issues suggest a more controversial contribution of XCI in Choroideremia pathogenesis [18]. First, some studies showed that XCI seems to not correlate with disease manifestations in females, with affected females showing either a random X-inactivation pattern or a significant preferential inactivation of chromosome X carrying the mutant allele [19]. In our case, XCI studies on II:2 did not show a significant skewing of XCI in the blood sample (data not shown). Moreover, both the results in favor of and against the correlation between XCI and Choroideremia could be biased by the fact that, due to the inaccessibility of the retina, different types of tissues are utilized for such analysis. Since different tissues present with a different propensity for age dependent XCI skewing, XCI assay results could not be reliable for the retinal XCI ratio estimation, especially those obtained from blood, which, among others, has a high propensity for this phenomenon [19]. In addition, other studies have shown that XCI retinal patches do not correlate with regions of the greatest functional impairment, as shown by multifocal electroretinography, thus suggesting that XCI skewing is not sufficient to explain all the retinal perturbations occurring in female carriers [19]. For these reasons, other mechanisms could play a role in Choroideremia manifestation among females.

Interestingly, some evidence from the literature showed that the *CHM* gene could escape the XCI. One suggestion came in 1999 from Carrel L and Williard HF, who demonstrated that, in six out of nine cultured fibroblast cell lines from females with complete nonrandom XCI, REP1 presented with biallelic expression, either from the X-active (Xa) or from the X-inactive (Xi) [20]. Afterwards, in 2005, *CHM* expression from Xi was demonstrated again in six out of nine hybrid rodent/human somatic cell hybrids [21]. Moreover, *CHM* presented with a female-biased expression, as for other genes known to escape XCI [22]. Overall, the current evidence from literature suggests that *CHM* expression could present with inter-individual difference, showing biallelic expression in some females and monoallelic expression in others [20,21,22]. However, these results are still not sufficient to conclude that biallelic expression of *CHM* could indeed have a role in Choroideremia pathogenesis, since, as for XCI, the related studies were not performed on retinal tissue. Even though the presence and the grade of biallelic expression in retinal cells of *CHM* still needs proper demonstration, this emerging complexity of *CHM* expression could mirror, at least partially, the variable manifestation of the disease in *CHM* female carriers. Thus, it is worth postulating a hypothetical model in which the biallelic expression of *CHM* would concur in the manifestations of Choroideremia in females.

In females carrying a *CHM* variant that abolishes the expression of one REP1 allele (i.e., deletion, frameshift, nonsense), the *CHM* mutated allele would not produce a final protein product; hence, Choroideremia manifestation would depend only upon XCI status of the wild type allele: the more cells presenting with inactivated wild type allele, the more involvement of the retina would occur. Conversely, in female carriers of in-frame variants impairing the binding of REP1 with Rab but not with RGGT, as in our case, because of XCI escaping, both the mutated and the wild type of *CHM* alleles will be expressed at variable levels resulting in the translation of two different REP1 isoforms. Following the alternative model (Figure 1), mutant REP1 would bind a portion of RGGTs, making them unavailable for prenylation of Rab proteins, therefore acting with a dominant-negative effect. The fact that this type of variant could be expressed in all retinal cells would indeed produce a greater global effect on retinal physiology, likely leading to more severe phenotypes in female carriers, such as those shown in our cases. Figure 7 summarizes the putative mechanisms considered. In order to corroborate the in silico model presented in this study and to strengthen the possible association between in-frame variants perturbing only REP1/Rab interaction and a worse phenotype in female carriers, we analyzed the distribution and potential structural impact on *CHM* of two other missense variants (c.101T>A,p.Val34Asp and c.940G>A,p.Gly314Arg) from two females reported by Jauregui et al., 2019 [4]. According to our model, the p.Val34Asp variant, which was reported to be associated with a severe phenotype, is located at the N-terminus, but nonetheless interacts with the C-terminus of the REP1 domain, where the p.Val34 residue is engaged in key hydrophobic interactions with the β strand 390–397, at the interface with Rab (Appendix A). Therefore, the p.Val34Asp variant affects the same interface region that is distorted in the c.1349+1G>A mutation and removes the committed conformation of *CHM* for Rab binding. As in our case, the p.Val34Asp mutant would still be able to interact with RGGT but would not be able to attach to Rab. On the contrary, the p.Gly314Arg variant, which was reportedly associated with a mild phenotype, is located on the surface of the C-terminus but does not affect Rab binding (Appendix A). More evidence from functional in vitro studies is needed to confirm the hypothetical model hereby proposed. Firstly, the predicted in silico effect of the *CHM* c.1349+1G>A needs to be demonstrated through expression of mutated and wild type *CHM* cDNAs in cell cultures, both to confirm its pathological effect and the true functional interactions between wt/ mutated REP1, Rab and RGGT. This could be achieved, respectively, through sub-cellular protein fractionation to demonstrate reduction in membrane-bound Rab and by co-immunoprecipitation assay to demonstrate that mutated REP1 would lose binding capacity with Rab while retaining that with RGGT. Secondly, our findings further stress the necessity, already discussed above, to find new ways to study the role of X inactivation and gene escaping in hereditary retinal disease that would not be biased by the different origin of the analyzed sample. One promising emerging method could be the use of retinal organoids differentiated form human induced pluripotent stem cells (hiPSC), which have been demonstrated to be a good proxy for replicating retinal human development and physiology [23]. The major advantage of this approach is the possibility of obtaining an in vivo model of an inherited retinal disease, one that, starting from a patient’s sample, represents, on the retinal tissue, not only the effect of the specific pathogenic variant, but also the entire genomic/epigenomic context in which the disease manifests.

## 5. Conclusions

The manifestation of Choroideremia in female carriers remains far from being fully elucidated. Unraveling the mechanisms underlying the manifestation of Choroideremia in female carriers is crucial for their correct clinical recognition and follow-up. The wide spectrum of reported phenotypes suggests that the degeneration of the retina could also occur in females but with a slower rate of progression with respect to males. The sex-related penetrance of *CHM* alterations may depend not only on the XCI skewing ratio between the mutant and wild type alleles but could also be influenced by the presence of a biallelic expression of *CHM*. Both of these phenomena remain to be confirmed on retinal tissue to prove their role in Choroideremia pathogenesis. In addition, if confirmed with further studies and in large cohorts of female carriers, biallelic expression of *CHM* could add another level of inter-individual variability among female carriers, opening up the possibility of specific genotype–phenotype correlations, especially in the presence of a *CHM* in-frame pathogenic variant. Between those variants, the ones predicted to impair REP1/Rab binding without alteration of REP1/RGGT interaction could produce a dominant negative effect by sequestrating RGGT, thus reducing its available amount for Rab prenylation. The last effect could have serious repercussions on the outcome of present therapeutic strategies for Choroideremia, in males, as well. In fact, transgene expression of wild type *CHM* copies through adeno-associated virus (AAV) vectors’ infection of retinal cells would not suffice in restoring physiological prenylation of Rab proteins due to competition between the mutated and the wild type transfected *CHM* alleles for RGGT binding. In those cases, therapeutic strategies should eventually also include methods for enhancing RGGT activity (i.e., through simultaneous transfection of wild type RGGT copies) or by inhibition of *CHM* mutated allele through antisense oligonucleotides (ASOs).

## Figures and Tables

**Figure 1 genes-13-01268-f001:**
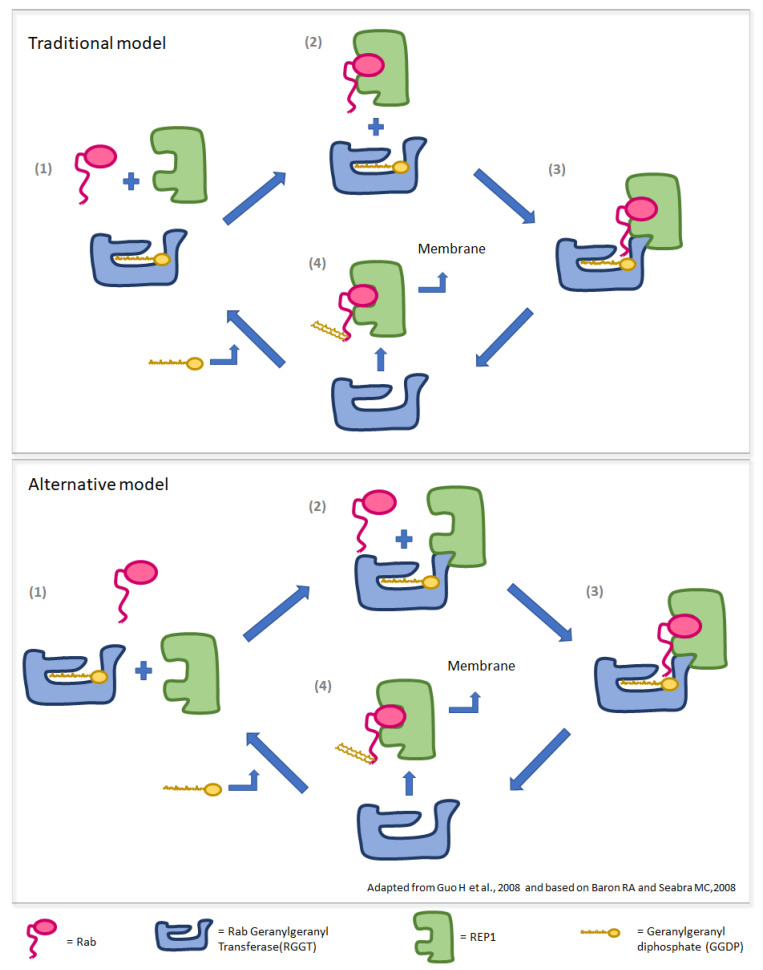
Models of REP1 mediated prenylation of Rab proteins. (Upper panel). Traditional model: (1) REP1 binds Rab; (2) Rab/REP1 complex binds RGGT/GGDP; (3) RGGT mediates two cycles of geranylation of Rab using two molecules of GGDP; (4) REP1 detaches from RGGT and escorts RabGG to membranes. (Lower panel). Alternative model: (1) REP1 binds RGGT/GGDP; (2) Rab binds REP1/RGGT/GGDP complex; (3) RGGT mediates two cycles of geranylation of Rab using two molecules of GGDP; (4) REP1 detaches from RGGT and escorts RabGG to membranes.

**Figure 2 genes-13-01268-f002:**
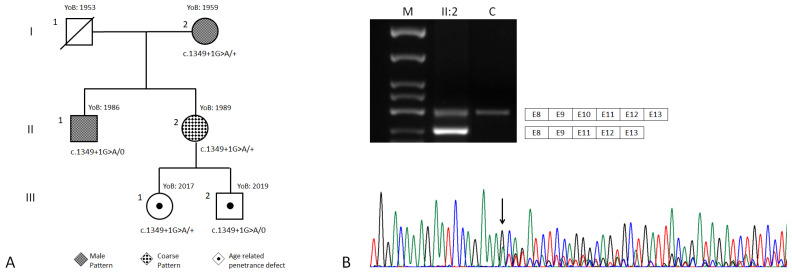
(**A**) Family pedigree. The phenotype of each patient is shown with different filling motifs. *CHM* genotypes are provided under each case’s symbol. The clinical legend is provided under the pedigree. YoB: year of birth. (**B**) cDNA analysis of cases II:2 and a control. The gel electrophoresis of the PCR products amplified with primers from exon 8 to exon 13 is shown on the top. The Sanger sequencing electropherogram from cDNA of case II:2 is shown on the bottom. M: molecular weight marker 50 (Experteam, Venezia, Italy).

**Figure 3 genes-13-01268-f003:**
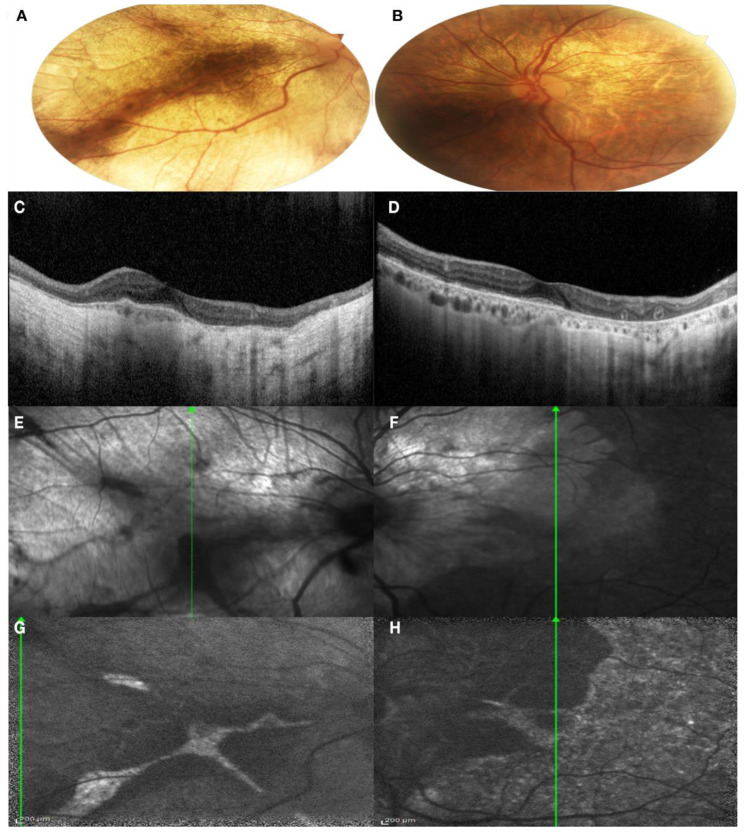
Multimodal imaging of both eyes of Patient I:2. (**A**,**B**) Color fundus photography of both eyes showed a large bilateral chorioretinal atrophy; (**C**,**D**) SD-OCT of both eyes showed an extensive chorioretinal atrophy and retinal thinning in the right eye and a residual central island of preserved Ellipsoid Zone (EZ) area in the left eye; (**E**,**F**) IR images showed areas of residual RPE in the left eye; (**G**,**H**) FAF showed generalized decreased autofluorescence, with residual areas of autofluorescence in the central macular area in the right eye and also in the temporal macular area in the left eye. (**A**,**C**,**E**,**G**): right eye; (**B**,**D**,**F**,**H**): left eye.

**Figure 4 genes-13-01268-f004:**
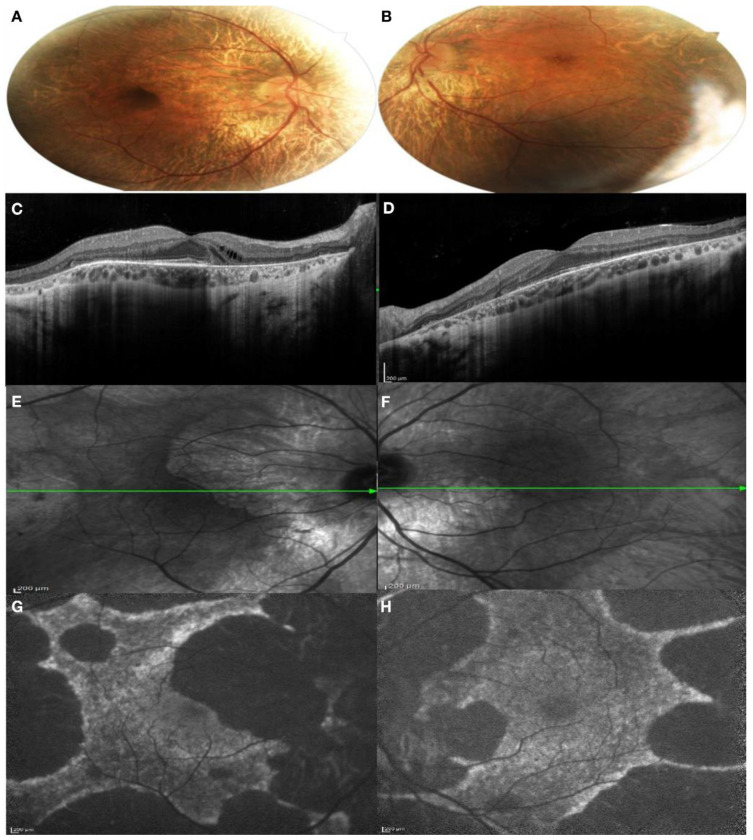
Multimodal imaging of both eyes of patient II:1. (**A**,**B**) Color fundus photography of both eyes showed large peripapillary and temporal retinal atrophy; (**C**,**D**) SD-OCT of both eyes showed peripapillary chorioretinal atrophy with retinal thinning and hyper-transmission posterior to the RPE. In the foveal and iuxtafoveal area, the Ellipsoid zone (EZ), RPE and all retinal layers were well-detectable; (**E**,**F**) IR images showed the transition of the intact RPE under the fovea to the atrophic area; (**G**,**H**). FAF showed central island of preserved autofluorescence. (**A**,**C**,**E**,**G**): right eye; (**B**,**D**,**F**,**H**): left eye.

**Figure 5 genes-13-01268-f005:**
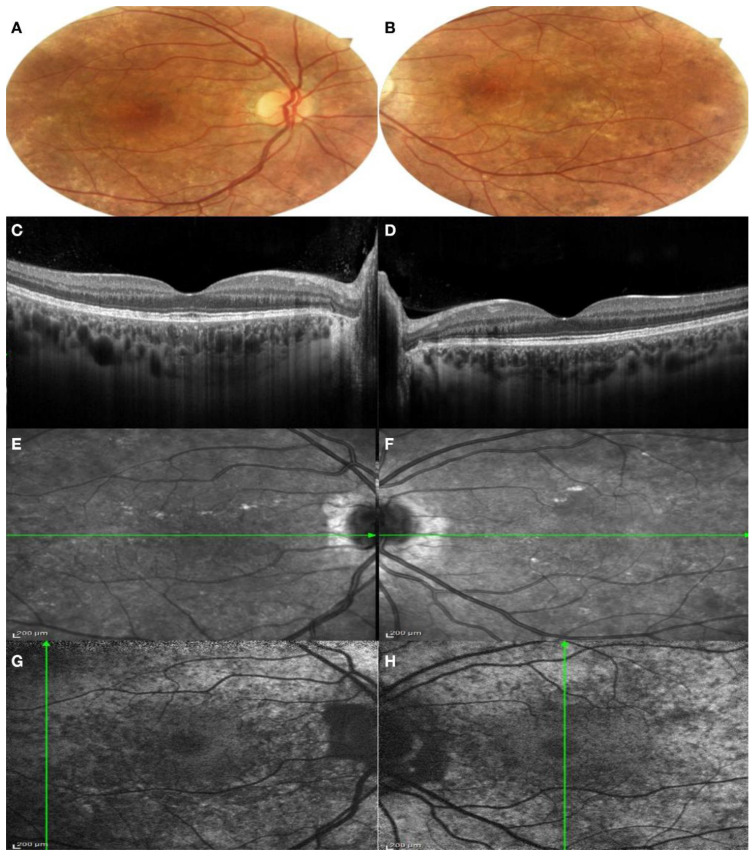
Multimodal imaging of both eyes of Patient II:2. (**A**,**B**) Color fundus photography of both eyes showed mild peripapillary atrophy, areas of hyper-pigmentation in the perifoveal area and pigmentary changes at the level of the vascular arcades; (**C**,**D**) SD-OCT of both eyes showed thinning of the ONL in the perimacular zone. IZ-EZ and the RPE–Bruch’s complex were irregular, with spot of hyper-transmission posterior to the RPE-Bruch’s membrane; (**E**,**F**) IR images showed normal aspect of the macular area and hypo-reflective areas among the vascular arcades; (**G**,**H**) normal autofluorescence of the macular area and a generalized hypo-fluorescence of the vascular arcade area. (**A**,**C**,**E**,**G**): right eye; (**B**,**D**,**F**,**H**): left eye.

**Figure 6 genes-13-01268-f006:**
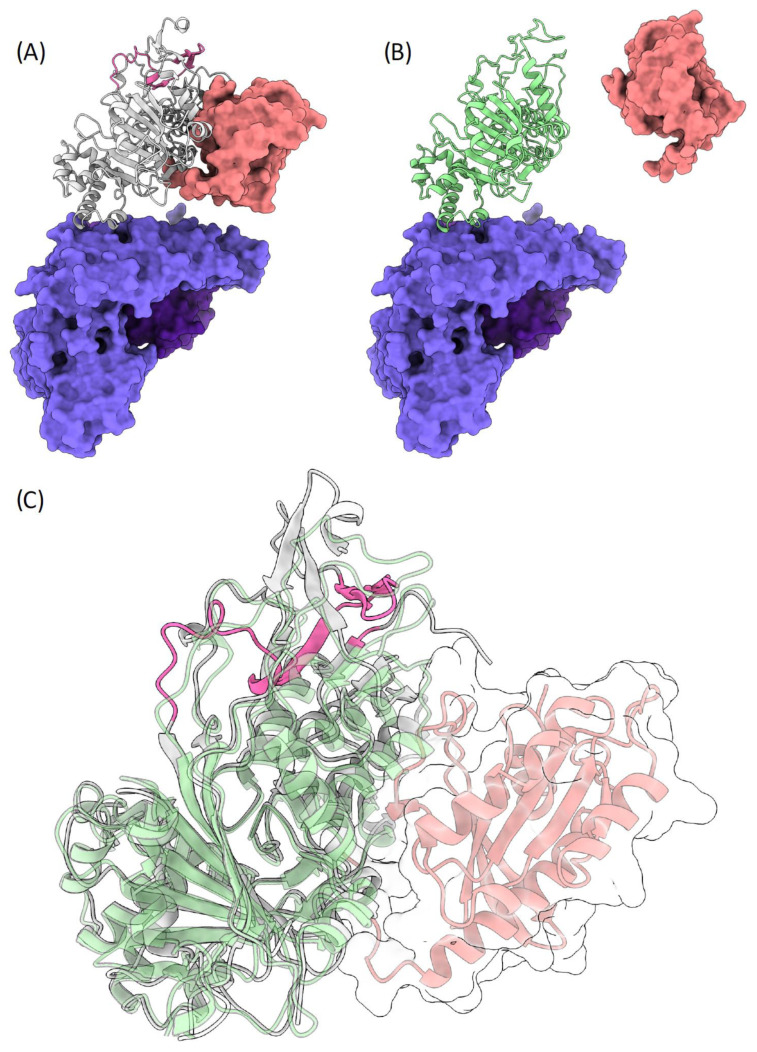
(**A**) The structure of wild type REP1 is shown in grey, with the peptide codified by exon 10 colored in magenta. The interacting RGGT is shown as a deep purple surface, while Rab7 is shown as a pink surface. (**B**) The image shows the interaction lost due to the lack of exon 10 in mutated REP1 (green illustration), while the interaction with RGGT is presumably conserved, given the invariant interaction interface. (**C**) A zoomed view of the interaction with Rab7 (pink illustrations and transparent surface) showing the different interaction interface affecting the binding of Rab7 due to the different conformation of REP1, lacking exon 10 (magenta).

**Figure 7 genes-13-01268-f007:**
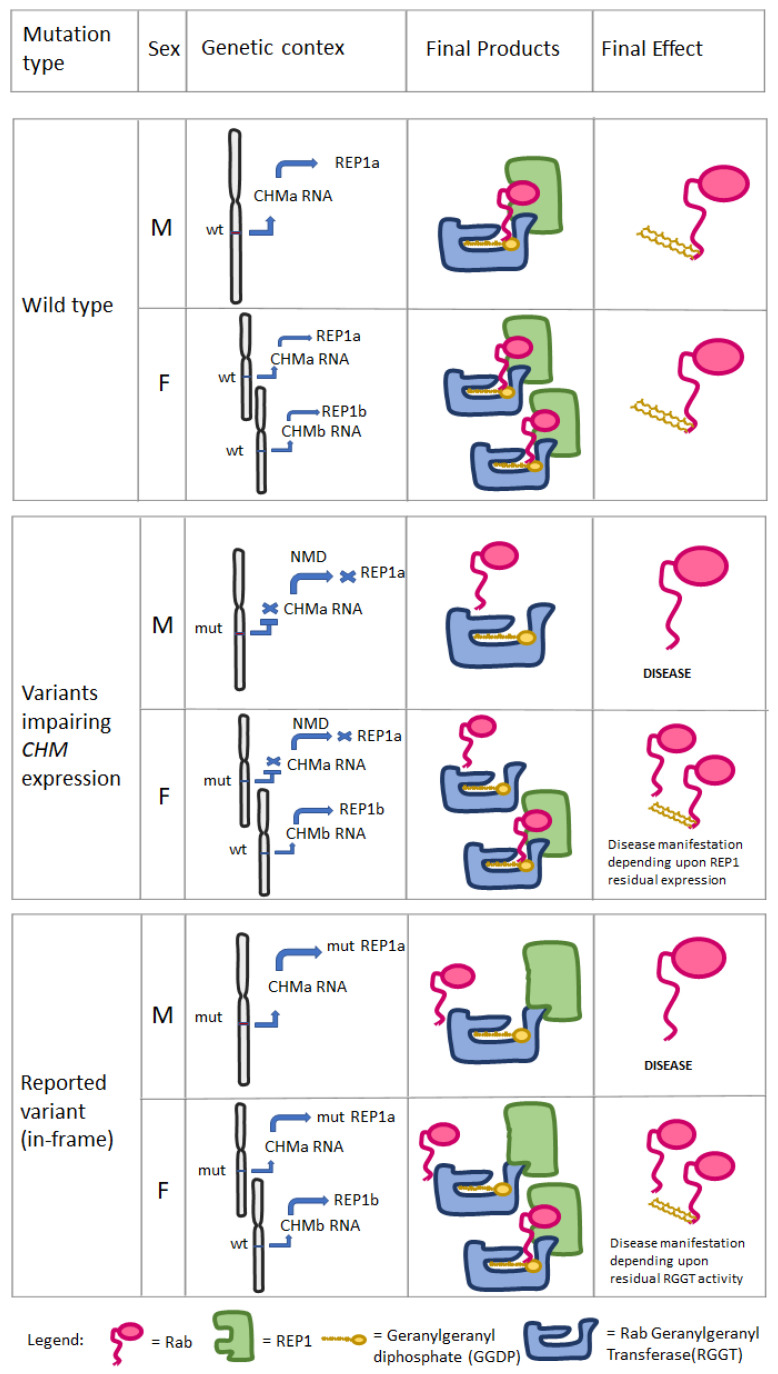
Proposed pathogenic molecular mechanisms for Choroideremia considering *CHM* biallelic (“a” allele and “b” allele) expression in females, following the alternative model (see Figure 1). The level of REP1a and REP1b expression would depend upon *CHM*a/*CHM*b position on Xa or Xi and the levels of expression of *CHM* allele from Xi. Legend: wt = wild type allele, mut = mutant allele, NMD= Nonsense RNA Mediated Decay.

## Data Availability

The data presented in this study are available on request from the corresponding author.

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
