# Peer review of "A Novel Hypothesis on Choroideremia-Manifesting Female Carriers: Could CHM In-Frame Variants Exert a Dominant Negative Effect? A Case Report"

_genes, 2022, doi:10.3390/genes13071268_

Round 1

Reviewer 1 Report

Brief Summary 

The interesting manuscript by Giosaffatte et al. describes detailed clinical findings of Choroideremia in a family segregating for the CHM c.1349+G>A variant.  The female 63 year-old proband, her son and her daughter were all shown by multimodal imaging to have severe anomalies in retinal, RPE and choroidal structures, consistent with a diagnosis of Choroideremia. cDNA analysis of a peripheral blood sample from one of the affected females indicated that the variant allele produces a REP protein lacking exon 10. In silico modelling was then used to predict possible perturbations in the binding interaction between REP, Rab7 and RGGT caused by variant REP. From this analysis the authors propose that the CHM c.1349+G>A variant may cause a dominant negative affect over wild-type REP via sequestration of RGGT.

General Comments

On the whole, this manuscript is well structured and relevant, however clarity of meaning could be improved (see Specific comments below). 

In addition, although perhaps beyond the scope of this particular report, there should at least be a discussion of future biochemical analyses which would provide experimental support for the in silico findings. Specifically, expression of different combinations of cDNAs encoding wild-type and/or variant REP and Rab7 and RGGT proteins in cell culture followed by immunoprecipitation studies could elucidate true functional interactions between the 4 proteins. Furthermore, sub-cellular protein fractionation from similar protocols may be able to show whether there is an associated reduction in membrane-bound Rab due to expression of the variant REP protein.

In silico modelling of other established CHM variant(s) associated or un-associated with female carrier Choroderemia manifestation may also strengthen the in silico results presented here and are highly recommended.

The last paragraph of the Discussion is very unclear. Is the first sentence necessary? Please re-write this.

The authors should discuss the serious repercussions pertaining to gene therapy if the variant causes a dominant negative disease, i.e. “simple” CHM gene replacement would not suffice and either RGGT would also need to be expressed in the therapy and/or variant CHM allele/transcript would have to be suppressed.

In the Methods Section 2.3. PCR methods and primers are not given.

Is Figure 1 adapted from Guo 2008 (Ref 6) or Guo 2015 (not cited)?

In Figure 2A the genotypes for III-1 and -2 are the wrong way around.

Figure legends for Figures 3-5 should state whether A,C,E,G and B,D,F,H are right or left eyes.

Specific comments 

Line31-32 Change to: “mRNA studies revealed a shorter in-frame CHM isoform lacking exon 10.”

Line 103:  Reference 16 is cited first before Ref (8). 

Line 116: “blood” has been left out.

Line 202 Change to: “Exon 10 spans 105 nucleotides so that its skipping due to the presence of the c.1349+G>A allele should not alter the reading frame of CHM mRNA or create a truncated transcript.”

Line 208 Change to : “…functional studies (Section 3.3 below) supportive…”

Line 316-7 Change to: “ Figure 7 summarizes the putative mechanisms considered.”

Author Response

Reviewer 1

Brief Summary 

The interesting manuscript by Giosaffatte et al. describes detailed clinical findings of Choroideremia in a family segregating for the CHM c.1349+G>A variant.  The female 63 year-old proband, her son and her daughter were all shown by multimodal imaging to have severe anomalies in retinal, RPE and choroidal structures, consistent with a diagnosis of Choroideremia. cDNA analysis of a peripheral blood sample from one of the affected females indicated that the variant allele produces a REP protein lacking exon 10. In silico modelling was then used to predict possible perturbations in the binding interaction between REP, Rab7 and RGGT caused by variant REP. From this analysis the authors propose that the CHM c.1349+G>A variant may cause a dominant negative affect over wild-type REP via sequestration of RGGT.

General Comments

On the whole, this manuscript is well structured and relevant, however clarity of meaning could be improved (see Specific comments below).

Task 1: In addition, although perhaps beyond the scope of this particular report, there should at least be a discussion of future biochemical analyses which would provide experimental support for the in silico findings. Specifically, expression of different combinations of cDNAs encoding wild-type and/or variant REP and Rab7 and RGGT proteins in cell culture followed by immunoprecipitation studies could elucidate true functional interactions between the 4 proteins. Furthermore, sub-cellular protein fractionation from similar protocols may be able to show whether there is an associated reduction in membrane-bound Rab due to expression of the variant REP protein.

Reply: We thank the reviewer for the suggestion. In the discussion session, we added a paragraph about possible future biochemical analyses for supporting the in silico effect of the variant and to investigate the role of XCI and CHM gene escaping. We hope the reviewer will be pleased by this last addition.

Task 2: In silico modelling of other established CHM variant(s) associated or un-associated with female carrier Choroderemia manifestation may also strengthen the in silico results presented here and are highly recommended.

Reply:

We wish to express our gratitude to the referee for this insightful suggestion. Indeed, in order to corroborate the in silico model presented in the manuscript, we analysed the distribution and potential structural impact on CHM of two other variants from Jauregui et al., 2019, c.101T>A,p.Val34Asp and c.940G>A,p.Gly314Arg:

CHM variant

Reference

Patient ID

Sex

Status of the female carrier

c.101T>A,p.Val34Asp

Jauregui et al., 2019

2

F

severe

c.940G>A,p.Gly314Arg

Jauregui et al., 2019

10

F

mild

According to our model, the Val34Asp variant, which is associated to a severe status, is located at the N-terminus, but nonetheless interacts with the C-terminus of the REP-1 domain, where the Val residue is engaged in key hydrophobic interactions with the β strand 390-397, at the interface with Rab (Supplementary Figure 1A). Therefore, the Val34Asp variant affects the same interface region that is distorted in the c.1349+1G>A mutation, and removes the committed conformation of CHM for Rab binding. The Val34Asp mutant would still be able to interact with GGTA, but not be able to attach to Rab. On the contrary, the Gly314Arg variant, which is associated to a mild status, is located on the surface of the C-terminus, but does not affect Rab binding, and results in a milder condition (Supplementary Figure 1B). This part was added in the discussion session, following the proposed molecular model for our variant

Task 3: The last paragraph of the Discussion is very unclear. Is the first sentence necessary? Please re-write this.

Reply: The last paragraph of the discussion section was re-written following the reviewer comment. We hope that the new form will be clearer

Task 4: The authors should discuss the serious repercussions pertaining to gene therapy if the variant causes a dominant negative disease, i.e. “simple” CHM gene replacement would not suffice and either RGGT would also need to be expressed in the therapy and/or variant CHM allele/transcript would have to be suppressed.

Reply: We thank the reviewer for the suggestion; we added a paragraph in the conclusion session.

Task 5: In the Methods Section 2.3. PCR methods and primers are not given.

Reply: PCR methods and primers were added to the methods section.

Task 6: Is Figure 1 adapted from Guo 2008 (Ref 6) or Guo 2015 (not cited)?

Reply: We apologize for the mistake. The reference in Figure 1 was corrected according to the reviewer’s comment

Task 7: In Figure 2A the genotypes for III-1 and -2 are the wrong way around.

Reply: We apologize for the mistake. The Figure 2 was corrected according to the reviewer’s comment

Task 8: Figure legends for Figures 3-5 should state whether A, C, E, G and B, D, F, H are right or left eyes.

Reply: we changed the Figures 3-5 legends according to the reviewer’s comment

Specific comments 

Task 9: Line31-32 Change to: “mRNA studies revealed a shorter in-frame CHM isoform lacking exon 10.”

Reply: the text was changed according to the reviewer’s comment

Task 10: Line 103:  Reference 16 is cited first before Ref (8).

Reply: we apologize for the mistake. We adjusted the citation.

Task 11: Line 116: “blood” has been left out.

Reply: the text was changed according to the reviewer’s comment

Task 12: Line 202 Change to: “Exon 10 spans 105 nucleotides so that its skipping due to the presence of the c.1349+G>A allele should not alter the reading frame of CHM mRNA or create a truncated transcript.”

Reply: the text was changed according to the reviewer’s comment

Task 13: Line 208 Change to : “…functional studies (Section 3.3 below) supportive…”

Reply: the text was changed according to the reviewer’s comment

Task 14: Line 316-7 Change to: “ Figure 7 summarizes the putative mechanisms considered.”

Reply: the text was changed according to the reviewer’s comment

Reviewer 2 Report

The authors report an interesting work that contributes to understand the molecular effects of a novel putative pathogenic variant of the CHM gene. Choroideremia is an X-linked retinopathy caused by mutations in the CHM gene. The pathogenic CHM variants CHM lead to progressive choroidal and retinal degeneration and eventually cause blindness.

CHM encodes REP1 (Rab escort protein) that plays an essential role in the cytosolic chemical modification (prenylation) of Rab. The complex Rab-REP1-RGGT (RGGT, Rab geranylgeranyl transferase) anchors in the membranes of intracellular organelles and vesicles and it regulates intracellular membrane trafficking. Two alternative pathways have been proposed to generate the Rab-REP1-RGGT complex.

After whole exome sequencing (WES) the authors report a putative CHM variant (CHM c.1349+1G>A) identified in a female patient (63 yo) segregating in the daughter and son. They also show by an in vivo assay that compares de CHM transcriptional products from exon 8 to exon 13 that the molecular effect of this mutation is an in-frame deletion of exon 10. CHM is an X-linked gene and female mutation carriers are mainly asymptomatic. However, this is not the case in the proband because after comprehensive ophthalmological evaluation the traits of a severe phenotype are clearly shown.

The authors go on trying to get structural insights into the tertiary structure and feasibility of REP1 complex with wild type and the putative mutant form. Based on the available crystal structure of REP1 and some of its complexes and using suitable templates. They succeed modelling the human ternary complex REP1/Rab7/GGTA and show that REP1 exon 10 (the peptide encoded is located at the C-terminus) deleted variant cannot interact with Rab, while the interaction with RGGT is still possible. This model, which still awaits experimental information, nicely shows that REP1 deficiency could affect the availability of RGGT for Rab geranylgeranyl modification, with a clear impact on the cellular function of the complex.

The authors try to find an explanation for the severe phenotype of the proband, and the differences in the clinical traits observed in both eyes . It could be hypothesized that both features are due to skewed X-Chromosome Inactivation (XCI). However, the authors put a lot of emphasis on reported data mainly obtained from other tissues (blood, cultured fibroblasts) to contravene the XCI hypothesis. It is true that some genes escape X-inactivation and are expressed from both the active and inactive X chromosome, but solid information on that point concerning CHM gene is missing.  Then, the discussion on this point (discussion section) should be tuned-down and considerably reduced. Accordingly, Fig 7 (proposed molecular mechanism for choroideremia considering CHM biallelic expression in females) should be omitted.

Author Response

Reviewer 2

The authors report an interesting work that contributes to understand the molecular effects of a novel putative pathogenic variant of the CHM gene. Choroideremia is an X-linked retinopathy caused by mutations in the CHM gene. The pathogenic CHM variants CHM lead to progressive choroidal and retinal degeneration and eventually cause blindness.

CHM encodes REP1 (Rab escort protein) that plays an essential role in the cytosolic chemical modification (prenylation) of Rab. The complex Rab-REP1-RGGT (RGGT, Rab geranylgeranyl transferase) anchors in the membranes of intracellular organelles and vesicles and it regulates intracellular membrane trafficking. Two alternative pathways have been proposed to generate the Rab-REP1-RGGT complex.

After whole exome sequencing (WES) the authors report a putative CHM variant (CHM c.1349+1G>A) identified in a female patient (63 yo) segregating in the daughter and son. They also show by an in vivo assay that compares de CHM transcriptional products from exon 8 to exon 13 that the molecular effect of this mutation is an in-frame deletion of exon 10. CHM is an X-linked gene and female mutation carriers are mainly asymptomatic. However, this is not the case in the proband because after comprehensive ophthalmological evaluation the traits of a severe phenotype are clearly shown.

The authors go on trying to get structural insights into the tertiary structure and feasibility of REP1 complex with wild type and the putative mutant form. Based on the available crystal structure of REP1 and some of its complexes and using suitable templates. They succeed modelling the human ternary complex REP1/Rab7/GGTA and show that REP1 exon 10 (the peptide encoded is located at the C-terminus) deleted variant cannot interact with Rab, while the interaction with RGGT is still possible. This model, which still awaits experimental information, nicely shows that REP1 deficiency could affect the availability of RGGT for Rab geranylgeranyl modification, with a clear impact on the cellular function of the complex.

Task 1: The authors try to find an explanation for the severe phenotype of the proband, and the differences in the clinical traits observed in both eyes. It could be hypothesized that both features are due to skewed X- Chromosome Inactivation (XCI). However, the authors put a lot of emphasis on reported data mainly obtained from other tissues (blood, cultured fibroblasts) to contravene the XCI hypothesis. It is true that some genes escape X inactivation and are expressed from both the active and inactive X chromosome, but solid information on that point concerning CHM gene is missing. Then, the discussion on this point (discussion section) should be tuned-down and considerably reduced. Accordingly, Fig 7 (proposed molecular mechanism for choroideremia considering CHM biallelic expression in females) should be omitted.

Reply: We thank the reviewer for the pertinent observation made. Actually, we think that the current evidence about both XCI skewing and CHM gene XCI escaping are still equally limited by the absence of clear and definitive studies performed on retinal tissue. We do not exclude XCI skewing as a pathogenic mechanism in females but some critical evidence in literature suggest that this mechanism is not sufficient for explaining their whole phenotypic spectrum. Indeed, it cannot be excluded that both mechanisms could play a modifier effect, with variable grade, in determining the phenotype of a female carrying a specific CHM pathogenic variant. For this reason, also in consideration of the predicted dominant negative effect of the CHM variant we found, we integrated both in a unique putative model responsible of the phenotype of our female carriers. We strongly believe that our model needs to be demonstrated by functional studies performed on retinal tissue or, as we added also following suggestions from the other reviewer, in retinal organoids. In consideration of the relatively novelty of the technique, we considered important to share our observation with the scientific community in order to stimulate further experimental studies to improve our understandings of choroideremia pathogenesis and to reveal putative genotype-phenotype correlation that could drive the therapeutic approach on the patients.

Accordingly, we revised the discussion section but we would prefer to maintain the figure 7 and the discussion part concerning CHM XCI escaping.